# Motor Competence between Children with and without Additional Learning Needs: A Cross-Sectional Population-Level Study

**DOI:** 10.3390/children10091537

**Published:** 2023-09-11

**Authors:** Amie B. Richards, Harriet G. Barker, Emily Williams, Nils Swindell, Kelly A. Mackintosh, Richard Tyler, Lucy J. Griffiths, Lawrence Foweather, Gareth Stratton

**Affiliations:** 1Applied Sports, Technology, Exercise and Medicine (A-STEM) Research Centre, Swansea University, Swansea SA1 8EN, UK; amie.richards@swansea.ac.uk (A.B.R.); 972123@swansea.ac.uk (H.G.B.); 907358@swansea.ac.uk (E.W.); n.j.swindell@swansea.ac.uk (N.S.); k.mackintosh@swansea.ac.uk (K.A.M.); 2Health Research Institute, Movement Behaviours, Health, and Wellbeing Research Group, Department of Sport and Physical Activity, Edge Hill University, Ormskirk L39 4QP, UK; tylerr@edgehill.ac.uk; 3Population Data Science, Swansea University Medical School, Swansea SA2 8PP, UK; lucy.griffiths@swansea.ac.uk; 4Physical Activity Exchange, Research Institute for Sport and Exercise Sciences, Liverpool John Moores University, Liverpool L3 2EX, UK; l.foweather@ljmu.ac.uk

**Keywords:** motor competence, children, youth, special educational needs, SEND, additional learning needs, ALN, Dragon Challenge

## Abstract

The aim of this study was to examine associations in motor competence between children with additional learning needs (ALN) and typically developing children. This cross-sectional study involved a nationally representative cohort of 4555 children (48.98% boys; 11.35 ± 0.65 years) from sixty-five schools across Wales (UK). Demographic data were collected from schools, and children were assessed using the Dragon Challenge assessment of motor competence, which consists of nine tasks completed in a timed circuit. A multi-nominal multi-level model with random intercept was fitted to explore the proficiency between children with ALN and those without. In all nine motor competence tasks, typically developing children demonstrated higher levels of proficiency than their peers with ALN, with these associations evident after accounting for age, sex, ethnicity, and socioeconomic status. This study highlights motor competence inequalities at a population level and emphasises the need for policymakers, practitioners, and researchers to prioritise motor competence development, particularly for children with ALN.

## 1. Introduction

In children, the physiological and psychosocial benefits of achieving sufficient levels of physical activity have been well documented [1]. However, in Wales, United Kingdom (UK), only 14% to 22% of children achieve the recommended levels of physical activity [2]. This is of particular concern given the positive association between physical activity and motor competence [3], defined as ‘a person’s ability to execute a wide range of motor acts in a proficient manner, including coordination of fine and gross motor skills that are necessary to manage everyday tasks, such as walking, running, jumping, catching, throwing, kicking, and rolling’ [3,4]. Motor competence is an important marker of children’s health and development, with research showing children with high motor competence accrue benefits on physical fitness, healthy weight status, bone density, executive functioning, and academic attainment, together with overall physical activity [3,5,6,7,8]. Moreover, there are long-term impacts that promoting motor competence has on future physical activity opportunities [3,9], as children who are more proficient in their motor competence are associated with having a higher quality of life as adults [7]. It is therefore important to develop motor competence in children to promote immediate health and well-being that can track into adulthood [10].

Given the importance of motor competence for children’s health, well-being, and development, it is surprising that few studies have reported national levels of directly measured motor competence [11]. Surveillance approaches are essential to track trends, identify inequalities, and for developing effective approaches to enhance children’s motor competence. As a result, recent research has focussed on the development of motor competence assessments that can be implemented at scale [12]. Common assessment methods include tests which assess discrete skills in isolation, such as the Test of Gross Motor Development (TGMD-3; [13]) and Movement-ABC (M-ABC; [14]). More recently, dynamic assessments have been validated, such as the Dragon Challenge [15], the Canadian Agility and Movement Skill Assessment of Physical Literacy (CAMSA; [16]), and the Athletic Skills Track [17]. The Dragon Challenge and CAMSA assess a range of combined and complex movement skills [18] through a continuous dynamic obstacle course and aim to provide a more authentic assessment environment to emulate the multi-skill and sports activities that are developmentally appropriate for older children and adolescents. Furthermore, these circuit-based measures enable the assessment of large groups of children in a short time, making them more feasible for collecting population-level data on motor competence [12,19].

Several studies have examined biological and demographic correlates of motor competence in children and adolescents. Age has been strongly positively correlated with motor competence [20,21], whilst body mass index (BMI) has been negatively correlated with motor competence [22,23]. Sex has shown inconsistent results [24], with some studies showing no differences [25,26], but others indicating that being a girl is correlated with stability skills [27,28] and being a boy correlated with object control and locomotor skills [21,29]. Ethnicity and socioeconomic status (SES) research in this area remains in its infancy. Yet, significant differences in motor competence have been reported between SES [30,31] in total, fine and gross motor competence [32], and in stability, but not object control [22]. Early research, from 2002, found no significant differences between ethnicities in motor competence [33], although more contemporary research, from 2018, has shown that children identifying with a South Asian ethnicity had poorer locomotor skills than children from a white or black ethnic background [34,35].

Another demographic group that may need targeted support for motor competence interventions concerns children with additional learning needs (ALN). In Wales, ALN is used as an umbrella term within the education system to describe any child who has a learning difficulty or disability that requires additional learning provision [36]. The term ALN recently replaced the label ‘special educational needs and disability’ (SEND) and aims to reflect a more holistic approach to supporting children with difficulties. According to the 2022 school census in Wales, 15.8% of the child population had ALN [37], which is in line with global figures of 15% of the world’s population experiencing some form of disability [38]. The Welsh Census [37] highlights that the most commonly reported type of additional learning needs are ‘speech, language and communication difficulties’ at 4.8%, followed closely by ‘behavioural, emotional and social difficulties’ at 4.6%, ‘general learning difficulties’ at 2.9%, ‘moderate learning difficulties’ at 2.6%, and ‘Autistic Spectrum Disorder’ (ASD) at 2.1%. To our knowledge, no studies have compared motor competence in children with additional learning needs (ALN) with typically developing children at a population level using an objective, dynamic measure.

To date, research exploring motor competence and disabilities has generally focussed on clinical populations, such as children with ASD [39], Attention Deficit Hyperactivity Disorder (ADHD; [40]), Down Syndrome [41], and CHARGE syndrome [42], concluding that children living with these health conditions have poorer fundamental movement skills than typically developing children. Indeed, a recent systematic review highlighted research on intellectual disabilities (ID) and fundamental movement skills within the child and adolescent population [43] and concluded that although children with ID showed deficits in their fundamental movement skills, there is a dearth of research on ALN and motor competence at a population level. Research to date is limited as studies on clinical populations have a low number of participants [44] and none have used dynamic assessment measures that are relevant for investigating motor competencies that are developmentally appropriate for older children. An example of this includes a study where the majority of children with ASD (82%) showed significant motor delays using the M-ABC assessment; however, this study only included fifty-one children, of which only five were girls [45]. Similarly, a systemic review investigating the impact of medication on ADHD children and their motor skills also cited its limitations as small sample sizes and a lack of female participants [46]. One population-level study did focus on the coordination element of motor competence between children with and without ADHD [11], finding significant differences over time. Children with ADHD had lower coordination performance than children without ADHD, and these differences persisted over the 11-year period. Nevertheless, population-level data comparing levels of motor competence among children with and without ALN are required to provide a deeper understanding of motor behaviours and abilities within this large proportion of the population.

Despite almost a sixth of children in Wales having an ALN [37], there remains a paucity of research within this large population. Studies investigating motor competence in children with ALN, in comparison to typically developing children, are urgently needed given the relationship between physical activity, motor competence, health, and well-being [3]. The aim of this study therefore was to explore motor competence in children with and without ALN at a population level, with the primary research question being whether typically developing children are more likely to be proficient in motor competence than children with ALN. Our initial hypothesis is that typically developing children will be more proficient than those with ALN.

## 2. Materials and Methods

### 2.1. Study Design and Participants

This study was of cross-sectional design and was conducted between November 2014 and November 2016. In total, 4555 children from 65 primary and secondary schools across all regions of Wales participated in the Dragon Challenge surveillance project to assess children’s motor competence. Children were invited to take part in the study and written consent was gained from parents and headteachers, and assent was gained from children. Ethics approval was granted in 2014 by the lead author’s institution (PG/2014/39).

### 2.2. Data Collection

#### 2.2.1. The Dragon Challenge

The Dragon Challenge is a valid, reliable, and dynamic measure of motor competence in children aged 10 to 14 years. All measures were conducted in accordance with Tyler et al. [15] and the Dragon Challenge manual, which is readily available online. Briefly, the Dragon Challenge comprises nine tasks, including three stability tasks (balance bench, core agility, wobble spot), three object control tasks (overarm throw, basketball dribble, underarm throw and catch), and three locomotor tasks (T-run, jumping pattern, and a sprint finish). Children watched a demonstration of the full Dragon Challenge and practiced each task in isolation before having a single attempt at the full Dragon Challenge. The Dragon Challenge is a hybrid-based assessment of motor competence that uses equally weighted processes (quality of technique), products (successfully achieve task goal), and time scores (time taken to complete circuit) to provide an overall score out of 56. This score is then given a category ranging from Bronze through to Platinum; more detail on these categories is available in the Dragon Challenge manual (https://cdn-links.lww.com/permalink/mss/b/mss_2018_07_25_tyler_18-00202_sdc3.pdf, accessed on 31 July 2023). All assessments were conducted in situ by trained assessors who received standardised training (at least five hours) to implement the Dragon Challenge assessment. Assessors also had no prior knowledge of the children’s movement capabilities or physical activity levels. The Dragon Challenge has previously shown good inter- and intra-rater reliability among assessors [15].

#### 2.2.2. Confounding Variables

Demographic characteristics, such as date of birth, sex, ethnicity, free school meal status (as a proxy measure of SES), and ALN were obtained from school demographic records. Date of birth was used to calculate the children’s decimal age, whilst ethnicities were categorised into ‘White’, ‘Asian’, ‘Black’, ‘Mixed’, and ‘Other’. There was no consistent sub-categorisation of ALN between the schools; some used categories such as ‘school action’, ‘school action +’, ‘statemented’, and ‘no provision’, whilst others simply used ‘yes’ or ‘no’. As such, this variable was collapsed into a dichotomous (yes or no) response.

### 2.3. Data Analysis

Of the 4555 participants, a total of 3489 participants (11.4 ± 0.6 years; 49.4% boys) from 55 schools were eligible for analyses. Data were excluded for participants missing school-level data (*n* = 175), demographic data (*n* = 750) or where there were errors, outliers, or missing data in the scores that were recorded (*n* = 123). Children whose recorded time was greater than four minutes (*n* = 18; 0.4%) were also removed from the analyses. Those participants whose data were excluded were similar to those used in the final analysis. The sampling error was 0.39%. Children were able to score zero, one, two, three, or four on each Dragon Challenge task. Each of the nine task scores were subsequently collapsed into two categories: ‘proficient’, which encompassed scoring a 4, and ‘not proficient’, which included scoring a 0, 1, 2, or 3. These were the dependent variables within the study.

Given that the data were collected across multiple schools, it was possible that children from the same school share similar motor competence profiles. Between-component variance, calculated using intraclass correlation coefficients, showed that schools accounted for between 4% and 23% of the variance in the dependent variables. Therefore, to account for the nested structure of the data, in conjunction with the proportional odds assumption not being met, a multi-nominal multi-level model, using Markov Chain Monte Carlo (MCMC) estimations with a random intercept, was fitted to investigate whether there were significant associations in motor competence between children of different ages, ethnicities, SES, and those with and without ALN. The use of MCMC estimations was due to the estimates being less biased and them being valid to apply to new models [47]. Six models were fitted sequentially using MLwiN (version 3.05; [48]). First, a single-level “null model” was fitted (model 1), followed by another “null model”, but this time with the individual and school-level structure fitted (model 2). The main variable of interest, ALN, together with covariates (decimal age, ethnicity, free school meal status, and sex) were then added (model 3), including a random intercept at the school level to account for the school-level variance. Models were also explored for ALN and decimal age interactions (model 4), ALN and free school meal interactions (model 5), and ALN and sex interactions (model 6). The Deviance Information Criteria (DIC) values were compared to ensure that the most appropriate model was used for analysis [49]. Model three has been reported as the final model as the interaction effects did not result in a significant improvement in the DIC values (see Appendix A). The alpha level for significance was set at *p* < 0.05 for all analyses and the Wald statistic was used within MLwiN to calculate significance. The regression coefficient was used to calculate odds ratios using the exponentiate function in Microsoft Excel. Coefficients, standard error, and credible intervals at 2.5% and 97.5% for all models were also reported (Appendix A).

## 3. Results

Descriptive data are included in Table 1. To summarise, the children had a mean age of 11.4 ± 0.6 years. Almost 94% of the children were of white ethnic origin, with the remaining 0.8%, 2.7%, 1.6%, and 1.1% being Black, Asian, Mixed, and other ethnicities. A fifth of children were classed as having ALN, whilst less than one in six were entitled to free school meals.

Table 2 shows means and standard deviations for tasks split by sex, ethnicity, ALN, and free school meal status. Using mean scores, girls scored higher than boys in balance bench, core agility, wobble spot, and jumping patterns. More affluent children scored higher than children entitled to free school meals in all tasks, apart from the overarm throw, where the opposite was true, and the basketball dribble, in which there were no differences between the groups. In all tasks, typically developing children performed better than their peers with ALN.

The DIC value for each regression model (1 through 3) decreased (Table 3). Therefore, the best fitting model, model 3, provided the main results in the form of odds ratios (OR) and significance values (Table 4). Results hereon in are discussed on a task-by-task basis.

### 3.1. Balance Bench

For the balance bench task, when accounting for age, ethnicity, free school meal status, and sex, typically developing children were 35% more likely to be proficient in the balance bench than their peers with ALN (OR = 1.35; *p* ≤ 0.01). Older children were significantly less likely to be proficient than younger children (OR = 0.92, *p* < 0.05), whilst Asian children were significantly less likely to be proficient than white children (OR = 0.65; *p* < 0.05). Girls were 19% more likely to be proficient than boys (OR = 1.19; *p* ≤ 0.01).

### 3.2. Core Stability

Similar to the balance bench task, when controlling for age, ethnicity, free school meal status, and sex, children who did not have ALN were 65% more likely to be proficient in the core stability task (OR = 1.65; *p* < 0.001) than their peers who had ALN. Older children were significantly more likely to be proficient than their younger peers (OR = 1.20, *p* < 0.001). Asian children were significantly less likely to be proficient than white children (OR = 0.43; *p* ≤ 0.01), whilst children not entitled to free school meals were significantly more likely to be proficient than less affluent children (OR = 1.40, *p* ≤ 0.01). Girls were significantly more likely to be proficient than boys (OR = 1.64; *p* < 0.001).

### 3.3. Wobble Spot

After accounting for age, ethnicity, sex, and free school meal status, typically developing children were 91% more likely to be proficient than children with ALN (OR = 1.91; *p* < 0.001). There were no significant sex or SES associations. For each year a child aged, they were 29% more likely to be proficient than younger children (OR = 1.29; *p* < 0.001). Asian children were significantly less likely to be proficient than white children (OR = 0.63, *p* < 0.05).

### 3.4. Overarm Throw

When controlling for age, ethnicity, sex, and free school meal status, typically developing children were 29% more likely to be proficient than children with ALN (OR = 1.29; *p* ≤ 0.01). The overarm throw task showed that older children were significantly more likely to be proficient than their younger peers (OR = 1.27; *p* < 0.001), whilst boys were also more likely to be proficient than girls (OR = 0.31; *p* < 0.001).

### 3.5. Basketball Dribble

Children with no additional needs were 49% more likely to be proficient than children with ALN (OR = 1.49; *p* < 0.001), after controlling for age, ethnicity, sex, and free school meal status. The basketball dribble was performed more proficiently in older children than their younger peers (OR = 1.35; *p* < 0.001). Asian children were less likely to be proficient than white children (OR = 0.50; *p* < 0.001) and girls were less likely to be proficient than boys (OR = 0.38; *p* < 0.001).

### 3.6. Underarm Throw and Catch

The model controlled for age, ethnicity, sex, and free school meal status and results revealed that typically developing children were more likely to be proficient than their peers with ALN (OR = 1.25; *p* ≤ 0.01). Older children were more likely to be proficient than younger children (OR = 1.16; *p* < 0.001), along with girls being less proficient than boys (OR = 0.43; *p* < 0.001).

### 3.7. T-Run

Typically developing children were 34% more likely to be proficient than children with ALN (OR = 1.34; *p* ≤ 0.01) when controlling for age, ethnicity, sex, and free school meal status, whilst girls were less likely to be proficient than boys (OR = 0.84; *p* < 0.05). Older children were more likely to be proficient than younger children (OR = 1.42; *p* < 0.001). Black children were 4.3 times more likely to be proficient than white children (OR = 4.38; *p* < 0.001), whilst children categorised as ‘other’ ethnicity were less likely to be proficient than white children (OR = 0.25; *p* ≤ 0.01). This was one of only four tasks where significant associations between SES were evident; children entitled to free school meals were 43% less likely to be proficient than their more affluent peers (OR = 1.43; *p* ≤ 0.01).

### 3.8. Jumping Patterns

When accounting for age, ethnicity, sex, and free school meal status, typically developing children were 58% more likely to be proficient than children with ALN (OR = 1.58; *p* < 0.001), while Asian children were less likely to be proficient in jumping patterns than white children (OR = 0.38; *p* < 0.001). Older children were significantly more likely to be proficient than younger children (OR = 1.07, *p* < 0.05), whilst children who were not entitled to free school meals were more likely to be proficient than children who were (OR = 1.19, *p* < 0.05).

### 3.9. Sprint Finish

Typically developing children were more likely to be proficient than children with ALN (OR = 1.47; *p* < 0.001) after controlling for age, ethnicity, sex, and free school meal status, whilst girls were less likely to be proficient than boys (OR = 0.65; *p* < 0.001). Older children were more likely to be proficient than younger children (OR = 1.17, *p* < 0.001), whilst Asian children were less likely to be proficient than white children (OR = 0.44; *p* < 0.001). More affluent children were 40% more likely to be proficient than children eligible for free school meals (OR = 1.40; *p* ≤ 0.001).

Although it was not the primary aim, we also looked at the interaction effects with age, sex, and SES (Appendix A). However, the inclusion of interaction effects did not significantly improve the model, suggesting that the effects of having ALN on motor competence are not significantly different across boys and girls, SES, and age.

## 4. Discussion

The purpose of this study was to ascertain whether there were population-level associations in motor competence between typically developing children and children with ALN using a dynamic assessment tool. We hypothesised that children with ALN would be less proficient in motor competence tasks than their typically developing peers.

When controlling for age, sex, ethnicity, and SES, typically developing children were more likely to be proficient in all Dragon Challenge tasks than children with ALN. Specifically, for balance and stability tasks, it could be postulated that such differences are evident due to the range of contexts where balance and stability skills can be developed, such as organised sporting opportunities and through habitual activity, unlike object control skills [50]. Indeed, recent evidence suggests that children with additional needs achieve less habitual physical activity such as running and jogging than typically developing children [51], which may, at least in part, reduce their proficiency in these skills.

Children with ALN were less proficient in object control skills than their typically developing peers; this is in accord with previous research which found that children with ID have poorer object control skills than their typically developing peers [52]. The object control skills incorporated within the Dragon Challenge can all be developed through exposure to organised sporting opportunities, such as basketball, cricket, or rounders. Congruent with previous research which suggested that the popularity of ball games for children and adolescents may play a part in the motor competence of object control skills [53], the basketball dribble was, on average, a high-scoring task. Recent data from the Active Lives Children Survey reinforce this, where team sports, the majority being ball games, were the most prevalent type of activity for 11–16-year-olds and the second most prevalent, only behind active play and informal activity, for 7–11-year-olds [54]. Having established that object control skills are developed through organised sporting opportunities and skilled instruction [50,55], it has been identified that children with ID, and presumably ALN, participate significantly less in organised sport than typically developing children [52]. This is also evident in Wales, where only 34.5% of children with any learning difficulty participate in sporting activities three times per week, compared to 41% of children with no learning difficulty [51]. Of interest, this same survey highlighted that for children with ALN, basketball was identified as the second most popular sport by demand, after swimming.

The last category of skills were the locomotor skills, consisting of a T-run, jumping patterns, and sprint finish. Typically developing children were significantly more likely to be proficient in these tasks than children with ALN. In this section of skills, there were the overall average lowest score (T-run, 2.00) and the overall average highest score (sprint finish, 3.42). However, again, children with ALN were less proficient at these locomotor tasks, consistent with previous research in children with ID [52] and learning disabilities [56].

Although the focus of this paper was on motor competence in children with and without ALN, there were also other between-group analyses that provide informative insights. The present study found that object-control skills were better in boys than girls, which is in accord with a previous meta-analysis [24]. Whilst boys outperformed girls in object control skills, the opposite was the case in stability skills, a finding which has previously produced equivocal results, with some studies consistent with this study [57] and others showing no differences [58]. Locomotor skills showed mixed results, with two tasks showing boys to be more proficient and one where girls were more proficient, which is consistent with other work [24]. Despite being a cross-sectional study, children’s age also affected performance, where older children were more proficient than the younger children in eight out of nine tasks, concordant with previous research [24]. Socioeconomic status and ethnic groups also performed differently. The present study revealed that more affluent children were significantly more likely to be proficient at four of the tasks than their peers who receive free school meals, similar to the findings of a previous study [59]. In two-thirds of the tasks, children who identified as having Asian ethnicity were less likely to be proficient than white children, in agreement with other studies [34,35].

A key message consistent across all tasks for children in any group is the level of exposure that they have to certain activities, and this is based on the opportunities provided. In Wales, only 9.6% of primary schools strongly agreed that staff members have enough support to engage with pupils with ALN when delivering physical education and sport [51]; this figure doubles to 20.8% in secondary schools; however, early exposure to appropriate movement and physical education is imperative for children to develop adequate levels of motor competence which can lead to lifelong health-promoting physical activity [60]. The exposure-to-activity theme continues where it is almost expected that boys will be more proficient in object control skills and girls in stability skills, as in secondary schools in Wales, where 21.2% of basketball sessions are male-only, compared to only 7.7% that are female-only. Conversely, gymnastics sessions are 5.2% and 11.0% male- and female-only, respectively.

Despite the lack of available data on children with ALN and motor competence, there are several intervention studies that have focussed on children with a range of disabilities. One systematic review summarised 14 motor skill interventions on fundamental movement skills in children with varying levels of ID [61], which all used a range of methods from Wii Fit training [62] for strength and balance activities [63] and adapted play training [64]. Despite significant improvements in balance or stability skills and in overall fundamental movement skills, these sample sizes were small. Interventions should be placed with larger ALN groups to assess any motor competence improvements with representative cohorts.

### Strengths and Limitations and Future Research

To our knowledge, this is the first study to provide a population-level analysis of children with ALN using a dynamic assessment of motor competence. The main strengths of this study include a large nationally representative sample size, across all four regions of Wales, whilst collecting demographic data, such as free school meals and age. Furthermore, this study used a multi-level analysis approach to account for the variation between schools and calculated odds ratios to indicate the practical implications of the results. Nevertheless, the following limitations are acknowledged. Odds ratios cannot be compared between studies and therefore they cannot be synthesised within a meta-analysis [65]. However, Appendix A provides the Beta coefficients that would allow meta-analyses to use the data. Socioeconomic status was measured through free school meal status, which is a crude measure compared to indices of multiple deprivation used in other studies [66,67]. Although motor competence is correlated with other variables in this study, physical activity, fitness, and motor competencies such as swimming and cycling were not directly measured and could have an impact on motor competence scores in the Dragon Challenge. Another limitation is that it is not evident as to what level of ALN the child has, and this could provide a greater depth of understanding within this study.

Considering the strengths and limitations, further studies should focus on providing nationwide, longitudinal insight into children’s motor competence. Such research would need a representative sample including children from multiple levels of deprivation, various ethnicities, and a reasonable proportion of children with ALN. Given the proportion of children who encompass the broad range of the ALN population, more focus should be given to this group, and intervention studies should not only focus on a clinical population but also on this broad ALN category to further reduce inequalities. Additional studies should also consider the role of process and product scores on children of varying abilities and how these scores could impact on any differences highlighted.

## 5. Conclusions

This research suggests that children with ALN require additional support and investment in improving motor competence, which should be made at an early age and in an inclusive setting, such as the school environment. Despite being a cross-sectional study, children’s scores increased with age, suggesting that any investments that are being made in children’s motor competence across time are effective, but not consistent among children with differing abilities, SES, and between boys and girls. Research and application in this area are gaining momentum, as a recent expert statement from the UK and Ireland emphasised these differences and provided recommendations for reducing these inequalities [5].

In conclusion, children’s motor competence at a population level is yet to be included in children’s health and well-being data; therefore, this study not only provides a basis but highlights that typically developing children are significantly more proficient than children with ALN. Policy makers should consider, and indeed develop, specific recommendations to provide the foundation for ensuring that inequalities in children’s motor competence are minimised and investments are made in addressing this gradient.

## Figures and Tables

**Table 1 children-10-01537-t001:** Participant characteristics.

Variable	All		Boys		Girls	
	Frequency	%	Frequency	%	Frequency	%
**Gender**			1722	49.4	1767	50.6
**Age**						
8 years	5	0.1	3	0.2	2	0.1
9 years	65	1.9	40	2.3	25	1.4
10 years	705	20.2	363	21.1	342	19.4
11 years	2342	67.1	1136	66.0	1206	68.3
≥12 years	372	10.7	180	10.5	192	10.9
**Ethnicity**						
White	3273	93.8	1615	93.8	1658	93.8
Black	27	0.8	12	0.7	15	0.8
Asian	94	2.7	41	2.4	53	3.0
Mixed	55	1.6	30	1.7	25	1.4
Other	40	1.1	24	1.4	16	0.9
**ALN**						
No	2784	79.8	1296	75.3	1488	84.2
Yes	705	20.2	426	24.7	279	15.8
**Free School Meals**						
No	2943	84.4	1442	83.7	1501	84.9
Yes	546	15.6	280	16.3	266	15.1

**Table 2 children-10-01537-t002:** The mean (M), standard deviation (SD), and percentage proficient (%) for each task, split by sex, ethnicity, free school meal status, and additional learning needs (ALN) status.

	Balance Bench			Core Agility			Wobble Spot			Overarm Throw			Basketball Dribble			Underarm Throw and Catch			T-Run			Jumping Patterns			Sprint Finish		
	M	SD	%	M	SD	%	M	SD	%	M	SD	%	M	SD	%	M	SD	%	M	SD	%	M	SD	%	M	SD	%
All (3489)	2.22	1.55	38.7	2.34	1.38	24.8	2.09	1.92	47.9	2.28	1.42	30.5	2.64	1.54	46.4	1.70	1.70	31.4	2.00	1.55	21.6	2.73	1.51	49.4	3.42	0.88	62.6
Boys (1722)	2.14	1.55	36.4	2.18	1.39	20.3	2.08	1.92	47.1	2.67	1.32	41.9	2.91	1.46	57.1	2.05	1.72	40.0	2.03	1.56	23.2	2.66	1.55	48.7	3.50	0.83	67.2
Girls (1767)	2.29	1.54	40.9	2.50	1.35	29.1	2.10	1.93	48.7	1.90	1.40	19.4	2.38	1.58	36.0	1.35	1.61	23.1	1.96	1.55	20.0	2.80	1.46	50.1	3.35	0.91	58.1
White (3273)	2.23	1.55	39.2	2.35	1.38	25.5	2.11	1.92	48.4	2.29	1.41	30.9	2.65	1.54	46.6	1.70	1.70	31.4	2.00	1.56	21.9	2.75	1.50	50.4	3.44	0.86	63.4
Black (27)	2.63	1.45	44.4	2.37	1.12	14.8	2.11	1.93	48.1	2.26	1.40	25.9	2.59	1.65	51.9	1.26	1.63	22.2	2.48	1.63	37.0	2.70	1.49	44.4	3.41	1.15	70.4
Asian (94)	1.84	1.53	29.8	2.04	1.32	12.8	1.52	1.89	35.1	2.05	1.34	21.3	2.27	1.63	35.1	1.61	1.72	30.9	1.79	1.42	10.6	1.96	1.49	23.4	2.93	1.08	38.3
Mixed (55)	2.05	1.52	30.9	2.47	1.20	18.2	1.82	1.98	43.6	2.13	1.52	30.9	2.80	1.60	54.5	1.95	1.70	36.4	2.24	1.54	25.5	2.91	1.32	47.3	3.33	0.86	52.7
Other (40)	1.88	1.49	27.5	2.10	1.26	15	1.90	1.95	42.5	2.13	1.44	25.0	2.65	1.44	42.5	1.70	1.71	32.5	1.58	1.39	7.5	2.58	1.50	42.5	3.33	1.05	60.0
Free School Meals = No (2943)	2.23	1.55	39.2	2.39	1.37	26.1	2.12	1.92	48.5	2.28	1.42	30.6	2.64	1.54	46.0	1.71	1.69	31.5	2.04	1.55	22.6	2.77	1.50	50.6	3.45	0.86	64.3
Free School Meals = Yes (546)	2.12	1.54	36.1	2.08	1.39	17.8	1.94	1.93	44.7	2.30	1.41	30.0	2.64	1.56	48.7	1.61	1.74	31.3	1.75	1.55	16.1	2.55	1.55	43.2	3.26	0.96	53.5
ALN = No (2784)	2.27	1.54	40.0	2.43	1.36	26.9	2.19	1.92	50.5	2.31	1.41	30.6	2.70	1.51	47.1	1.71	1.70	31.8	2.02	1.55	22.1	2.82	1.47	51.9	3.47	0.84	64.5
ALN = Yes (705)	2.02	1.58	33.5	2.01	1.39	16.3	1.69	1.89	37.4	2.20	1.44	29.9	2.42	1.64	43.7	1.65	1.69	29.9	1.89	1.55	19.9	2.37	1.60	39.6	3.23	1.00	55.2

**Table 3 children-10-01537-t003:** The Deviance Information Criteria (DIC) for each model built.

	Model 1	Model 2	Model 3	Model 4	Model 5	Model 6
Balance Bench	4658.836	4632.459	4619.820	4620.687	4617.389	4619.981
Core Agility	3910.001	3775.699	3715.943	3709.362	3714.668	3708.826
Wobble Spot	4832.631	4728.511	4678.374	4681.959	4681.005	4681.185
Overarm Throw	4293.612	4226.323	4011.409	4011.785	4015.564	4012.960
Basketball Dribble	4821.041	4721.006	4517.451	4517.538	4520.416	4518.480
Underarm Throw and Catch	4346.470	4311.087	4193.697	4191.132	4195.619	4194.462
T-Run	3644.146	3394.667	3375.390	3374.976	3373.000	3375.917
Jumping Patterns	4838.390	4625.221	4599.997	4600.737	4603.652	4601.199
Sprint Finish	4614.999	4384.170	4326.726	4329.427	4326.539	4329.327

**Table 4 children-10-01537-t004:** Model 3 odds ratios with significance levels indicated as follows: *p* ≤ 0.05 = ns, *p* ≤ 0.05 = a, *p* < 0.01 = b, *p* ≤ 0.001 = c.

	Balance Bench	Core Agility	Wobble Spot	Overarm Throw	Basketball Dribble	Underarm Throw and Catch	T-Run	Jumping Patterns	Sprint Finish
Decimal Age	0.92 a	1.20 c	1.29 c	1.27 c	1.35 c	1.16 c	1.42 c	1.07 a	1.17 c
Ethnicity_Asian	0.65 a	0.43 b	0.63 b	0.68	0.50 c	0.87	0.64	0.38 c	0.44 c
Ethnicity_Black	1.27	0.45	1.42	0.79	0.82	0.50	4.38 c	1.09	2.00
Ethnicity_Mixed	0.66	0.63	0.76	1.01	1.20	1.13	1.31	0.94	0.69
Ethnicity_Other	0.63	0.63	0.81	0.61	0.62	0.91	0.25 b	0.92	0.94
Free School Meal	1.11	1.40 b	1.03	1.05	0.86	1.02	1.43 b	1.19 a	1.40 b
ALN	1.35 c	1.65 c	1.91 c	1.29 b	1.49 c	1.25 b	1.34 b	1.58 c	1.47 c
Sex	1.19 b	1.64 c	1.03	0.31 c	0.38 c	0.43 c	0.84 a	1.07	0.65 c

Note: Reference is not proficient, reference ethnicity is white, reference free school meal is yes, reference ALN is yes, reference sex is boy.

## Data Availability

The data are available to the research team according to ethical approval. The corresponding author is happy to provide data if required for scrutiny.

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
