# Peer review of "Motor Competence between Children with and without Additional Learning Needs: A Cross-Sectional Population-Level Study"

_children, 2023, doi:10.3390/children10091537_

Round 1

Reviewer 1 Report

Dear authors, 

First of all I would like to congratulate you on this research. It is certainly very interesting. Before it is published, I would like the following changes to be made. 

Add the reliability analysis of the instruments used. Specifically Cronbach's Alpha. 

Then, at the end of the introduction, add the research question and the initial hypotheses of the study. 

Add the sampling error

Sample normality analysis has not been studied. This is a serious error

Author Response

Thank you for taking the time to review this manuscript. Please see the attachment.

Point 1:

Add the reliability analysis of the instruments used. Specifically Cronbach's Alpha.

Response 1:

In line 155 we refer to a paper on the reliability of the Dragon Challenge instrument. No change made to the manuscript.

Point 2:

Then, at the end of the introduction, add the research question and the initial hypotheses of the study.

Response 2:

I have added the following to the manuscript: The aim of this study therefore was to explore motor competence in children with and without ALN at a population-level; with the primary research question being whether typically developing children are more likely to be proficient in motor competence than children with ALN. Our initial hypothesis is that typically developing children will be more proficient than those with ALN.

Point 3:

Add the sampling error.

Response 3:

I have added the sampling error of 0.39% to the manuscript.

Point 4:

Sample normality analysis has not been studied. This is a serious error.

Response 4:

We did investigate normality and adjusted the methods accordingly. We chose to use the MCMC estimations as these do not require the normality assumption. No change made to the manuscript.

Reviewer 2 Report

Thank you for the opportunity to review this manuscript investigating motor competence between children with and without additional learning needs. The study was interesting and well written, and raises multiple population-level concerns relevant to student motor competence. Please consider the points of feedback below: 

Introduction

- Line 77: what is meant by early research, when was this? I believe that stating the timeline for this would be beneficial.

- Line 119: is there a reference for "Despite almost a sixth of children in wales having an ALN"?

Study Design and Participants

- Were the participants from primary/elementary school, or secondary/high school (or both)?

The Dragon Challenge

- Line 138: ((15), should be (15)?

- Line 147: what are the Bronze and Platinum categories? What exists in-between these two categories, and what is representative of each?

Data Analysis

- Lines 168-169: why were times greater than four minutes removed from the analyses?

- Line 176-177: how was the between component variance of 4% and 23% calculated?

Discussion

- Paragraph lines 410-421: there is quality information in this paragraph; I think it would be worthwhile to include this in the Introduction rather than leave to only the Discussion. This tells and unique story and provides important background.  

Author Response

Thank you for taking the time to review this manuscript. Please see the attachment. 

Point 1:

Line 77: what is meant by early research, when was this? I believe that stating the timeline for this would be beneficial.

Response 1:

ADDED TO MANUSCRIPT - from 2002 & 2018.

Point 2:

Line 119: is there a reference for "Despite almost a sixth of children in wales having an ALN"?

Response 2:

ADDED TO MANUSCRIPT - reference used previously in the introduction.

Point 3:

Study Design and Participants - Were the participants from primary/elementary school, or secondary/high school (or both)?

Response 3:

ADDED TO MANUSCRIPT - primary & secondary schools

Point 4:

Line 138: ((15), should be (15)?

Response 4:

EDITED IN MANUSCRIPT

Point 5:

Line 147: what are the Bronze and Platinum categories? What exists in-between these two categories, and what is representative of each?

Response 5:

ADDED TO MANUSCRIPT - more detail on these categories available in the Dragon Challenge manual.

Point 6:

Lines 168-169: why were times greater than four minutes removed from the analyses?

Response 6:

The Dragon Challenge development went through five iterations involving I've 200 practitioners in children's sport and PE from across all four regions of Wales. During the development phases child and school related factors were considered. To be able to assess efficiently children needed to complete the Dragon Challenge in half classes of between 10 and 15 children. The mean duration for DC completion was 2 minutes 29 seconds. Children who exceeded 4 mins generally lack focus on the tasks and it was deemed that running over a maximum time would not be in the best developmental interests for the children or be pragmatic in this study. Further work on developing a short form Dragon Challenge for very poor MC is in process to address this issue. No changes made to the manuscript.

Point 7:

Line 176-177: how was the between component variance of 4% and 23% calculated?

Response 7:

ADDED TO MANUSCRIPT - calculated using intraclass correlation coeffecients.

Point 8:

Paragraph lines 410-421: there is quality information in this paragraph; I think it would be worthwhile to include this in the Introduction rather than leave to only the Discussion. This tells and unique story and provides important background.  

Response 8:

Thank you for your suggestion, having reviewed this comment with co-authors we agree that we would like to leave these discussion points for the discussion as this is new data, we don't feel that this fits into the introduction.

Round 2

Reviewer 1 Report

The article has been improved. It can be published